# Survival and Demography of the Tomato Borer *(Tuta absoluta)* Exposed to *Citrus* Essential Oils and Major Compounds [†]

Gabriel Tadeu de Paiva Silva [1], Karolina Gomes Figueiredo [1], Dejane Santos Alves [2], Denilson Ferreira de Oliveira [3], Geraldo Humberto Silva [4], Gabriela Trindade de Souza e Silva [5], Murilo Silva de Oliveira [2], Antonio Biondi [6] and Geraldo Andrade Carvalho [1,*]

[1] Departamento de Entomologia, Universidade Federal de Lavras (UFLA), Lavras 7200-900, Minas Gerais, Brazil
[2] Campus Santa Helena (UTFPR-SH), Universidade Tecnológica Federal do Paraná, Santa Helena 85892-000, Paraná, Brazil
[3] Departamento de Química, Universidade Federal de Lavras (UFLA), Campus Universitário, Lavras 37200-900, Minas Gerais, Brazil
[4] Instituto de Ciências Exatas, Universidade Federal de Viçosa, Rio Parnaíba 38810-000, Minas Gerais, Brazil
[5] Departamento de Ciências Farmacêuticas, Universidade Estadual de Campinas, Campinas 13083-887, São Paulo, Brazil
[6] Department of Agriculture, Food and Environment, Universitá degli Studi di Catania, 95131 Catania, Italy
\* Correspondence: gacarval@den.ufla.br; Tel.: +55-3588179756
[†] This research work is the part of Master thesis of Gabriel Tadeu de Paiva Silva.

**Abstract:** *Tuta absoluta* is a pest of importance: quick to disperse and difficult to control due to the cases of resistance to insecticide active ingredients. Thus, studies using essential oils (EOs) to search for new molecules should be intensified. The objective of the present study was to evaluate the toxicity of EOs from *Citrus aurantifolia* (lime), *Citrus aurantium* (petitgrain) and *Citrus aurantium bergamia* (bergamot) and its major compounds against *T. absoluta* in a topical application test. Additionally, the demographic parameters of *T. absoluta* were studied after the topical application of EOs. The median lethal time ($LT_{50}$) of the population was 12h for the three EOs tested. The median lethal concentration ($LC_{50}$) was 33.75, 38.78 and 35.05 $\mu g \, \mu L^{-1}$ for *C. aurantifolia*, *C. aurantium* and *C. aurantium bergamia*, respectively. As found using gas chromatography coupled to mass spectrometry (GC-MS) quantification, 44.74% of the EO of *C. aurantifolia* is α-terpineol, while 55.45% and 58.12% of the EO of *C. aurantium* and *C. aurantium* bergamia, respectively, is linalyl acetate. The toxicity of the major compounds was tested at concentrations equivalent to the $LC_{50}$ of the EOs, that is, 16.2 $\mu g \, \mu L^{-1}$ for α-terpineol, and 25.8 $\mu g \, \mu L^{-1}$ for linalyl acetate, using topical application. Both of the major compounds showed less toxicity than the EOs. In the sublethal effects tests, all the EOs negatively affected the demographic parameters of *T. absoluta*, with a decrease in the duration of larval instars, duration of the pupal period, fecundity, oviposition and viability of the eggs, implying a reduction in the population growth parameters of this pest. The EOs of lime, petitgrain and bergamot are toxic to *T. absoluta*, and low concentrations cause deleterious effects on the reproductive and population parameters of *T. absoluta*.

**Keywords:** natural products; sublethal; botanical pesticides; Rutaceae; life table

## 1. Introduction

The tomato borer *Tuta absoluta* (Meyrick, 1917) (Lepidoptera: Gelechiidae) is a pest commonly found in plants of the Solanaceae family that can cause great economic losses when the larvae feed on leaves, stems, flowers and fruits [1–3]. Its life cycle is short, which increases its destructive potential, and after approximately 30 days, under an average temperature of 23 °C, a new generation of insects already occurs [1–3]. This pest is native to South America, but it took on a more prominent role in global terms from 2006, when it was detected in Spain and then spread quickly to other countries in Europe, Africa and Asia, reaching China, which is the world's largest tomato producer [2–6]. Furthermore,

*T. absoluta* is a major threat to other regions in Oceania and North America, since they have favorable climatic conditions for its survival and multiplication [7]. Models report that *T. absoluta* can spread by around 800 km per year and cause tomato production losses of 80–100% when the climate conditions are favorable [8,9].

Synthetic insecticides are the most commonly used method for the control of *T. absoluta*. However, there are numerous reports of the inefficiency of the active ingredients belonging to the chemical groups of pyrethroids, spinosyns, organophosphates, avermectins, cartap, indoxacarb, oxadiazines, diamides and benzoylureas due to their indiscriminate use, which has led to the selection of resistant populations of this pest [10–12]. Thus, studies looking for new molecules to control *T. absoluta* should be encouraged [13]. In this context, essential oils (EOs) have been studied for their control of *T. absoluta* due to their effectiveness, and the secondary metabolites present in EOs can be used as model molecules for the synthesis of new insecticides, such as pyrethroids that were synthesized from pyrethrin produced by chrysanthemum (*Tanacetum cinerariifolium*) [14]. The plants studied for the development of new molecules and new active essential oils generally show medical properties and are easy to produce to obtain and extract their essential oils [15–17]. These plants are known to contain active ingredients that are either toxic for or beneficial to humans [15–19]. In addition, the bioactive EOs can be used to formulate new biopesticides, such as Prev-Am®, which is mostly composed of *Citrus* peel oil and used to control and repel insects and mites [20].

The study of the acute toxicity of EOs is traditionally the most commonly used method for studying their lethal effect on insect pests; however, the sublethal effects of these chemicals are also important for toxicological studies since part of the population in the field is exposed only to low concentrations of the products [21]. The sublethal effects can increase the mortality rate of future generations and reduce reproductive parameters such as fertility and the period of oviposition, causing the reduction in the future populations of the target insect [21,22]. Thus, evaluating the impact of low concentrations of EOs on the life history of *T. absoluta* is important for understanding the toxicity of this insect [23].

For the study of sublethal effects and the life history of insects, fertility life tables are commonly used. However, these methods disregard the effect of males and the different stages of development in the life history of insects, leading to erroneous interpretations of the studied parameters [24,25]. To solve this problem, a method called the age-stage two-sex life table is proposed [26], where the survival and development of both males and females at all stages are considered. Thus, this methodology allows one to evaluate the effects of chemical molecules on demographic parameters that indicate the growth—or otherwise—of the insect population and reproductive parameters such as fertility, pre-oviposition and oviposition period, demonstrating effects on the population as a whole [27,28].

Considering the hypothesis that EOs negatively affect the biological characteristics of *T. absoluta* and that this toxic effect is caused by the major compounds of the mixture, the present study aimed to (i) evaluate the acute toxicity of EOs and the major compounds of *C. aurantifolia*, *C. aurantium* and *C. aurantium bergamia* against *T. absoluta* and (ii) study the effects of sublethal concentrations of EOs of *C. aurantifolia*, *C. aurantium* and *C. aurantium bergamia* on demographic parameters of *T. absoluta*.

## 2. Materials and Methods

### 2.1. Insects

The insects used in the bioassays were obtained for research rearing and maintained in the Laboratory of Ecotoxicology and Integrated Pest Management at the Universidade Federal de Lavras—Minas Gerais (UFLA-MG). The *T. absoluta* larvae were maintained in acrylic cages (60 × 30 × 30 cm) fed with tomato leaves from cv. Santa Clara, cultivated under greenhouse conditions (18–25 °C, 70% RU and air circulation) without any pesticide application. Newly emerged adults were transferred with a vacuum pump (EOS, model: VE290N-PRO) to another cage used for oviposition only, where tomato leaves were provided daily as a substrate and fed with 10% honey aqueous solution. After 24 h of egg laying, the

leaves were used in bioassays or transferred to new rearing cages (60 × 30 × 30 cm) and provided with tomato plants ad libitum. All bioassays and insect rearing were maintained at a temperature of 25 ± 2 °C, a relative humidity of 70 ± 10% and a 12L:12D photoperiod.

### 2.2. Essential Oils

The EOs of *C. aurantifolia*, *C. aurantium* and *C. aurantium bergamia*, exempt from impurities, were acquired from Ferquima Indústria e Comércio company, Vargem Grande Paulista, São Paulo, Brazil. The *C. aurantifolia* oil was obtained through the steam distillation of the essential oil extracted from cold-pressed fruits, the *C. aurantium* oil through the steam distillation of leaves and the *C. aurantium bergamia* from cold-pressed fruit peel.

### 2.3. Chemical Characterization of the EOs

The analyses were conducted using a gas chromatograph coupled to a mass spectrometer (model QP2010, Shimadzu, Japan) with an RTX-5MS capillary column (30 m × 0.25 mm ID × 0.25 μm film thickness; Restek). The EOs were diluted in acetone at a concentration of 10 mg mL$^{-1}$, with 1 μL of the solution injected into the gas chromatograph, in which helium was used at a rate of 1.0 mL min$^{-1}$ as the carrier. The conditions followed the one proposed by Adams [29], these being: split/splitless temperature: 220 °C; split injection ratio: 1:20; initial column temperature: 60 °C; rate of elevation in column temperature: 2 °C min$^{-1}$ to 200 °C, and after 200 °C, the rate of elevation changed to 5 °C min$^{-1}$; final column temperature: 250 °C; interface temperature between the chromatograph and the mass spectrophotometer: 220 °C; ionization of spectrophotometer molecules: electron impact at 70 eV; mass/load range (m/z) analyzed on the mass spectrophotometer: 45–400; mass spectrum acquisition time: 0.5 s. The components were identified based on comparisons with the relative retention index using data from a series of n-alkanes (C9-C20). All spectra were compared with a NIST 05 Mass Spectral Library 2005; peaks with a similarity of less than 90% were discarded.

### 2.4. Acute Toxicity of EOs against T. absoluta

The EOs of *C. aurantifolia*, *C. aurantium* and *C. aurantium bergamia* were diluted in acetone at a concentration of 100 μg μL$^{-1}$. From this solution, aliquots (1 μL) were applied to the dorsal surface of the larval thorax of each 2nd instar larva using a Hamilton® 50 μL microsyringe [19]. In the control treatment, the larvae were exposed only to acetone. After application, the larvae were individualized in wells of 32-cell white plastic rearing trays containing a piece of tomato leaf (3 cm × 3 cm) over a piece of filter paper (3 cm × 3 cm) that was previously moistened with 500 μL distilled water to maintain leaf tissue moisture and turgidity.

The bioassay was carried out in a completely randomized design, with four treatments and 60 repetitions, each formed by a 2nd instar larva, maintained individually in each well of the tray. The evaluation of the mortality of the larvae was made every 12 h up to 72 h after the application of the EOs. Larvae that did not show movements at the touch of a fine-tipped and soft-bristled brush were considered dead.

### 2.5. Time–Concentration–Mortality Responses of T. absoluta to the EOs

The EOs were diluted in acetone (17.3, 31.6, 41.3, 55, 75 and 100 μg μL$^{-1}$). The concentrations were determined using arithmetic progression and previous tests. *Tuta absoluta* 2° instar larvae were treated with 1 μL of the solutions using a microsyringe. In the negative control treatment, only the acetone solvent was used. The same experimental procedure as described in Section 2.4 was used in this test. The bioassay was carried out in a completely randomized design, with four treatments and 50 replicates, each formed by a larva kept individualized in each well of the plate. The evaluation of the mortality of the larvae was conducted every 12 h up to 72 h of their treatment, to determine the time–mortality response. The accumulated survival of insects after 72 h after the application of the treatments was used to calculate the concentration–mortality response.

### 2.6. Acute Toxicity of the Major Compounds of the EOs against T. absoluta

The major compounds of the EOs of *C. aurantifolia*, *C. aurantium* and *C. aurantium bergamia* (α-terpineol, 93% pure and linalyl acetate and 95% pure) were obtained from the Departamento de Química of Universidade Federal de Lavras (UFLA-MG). To calculate the concentrations of major compounds to be tested, the percentage of α-terpineol and linalyl acetate found in the CG-EM analysis was used, using the following formula:

$$CBT = CEO \times CMC(\%)$$

where *CBT* is the concentration to be tested; *CEO* is the concentration of EO and *CMC* is the content of the major compound present in the EO in percentage.

In this way, the treatments consisted of pure compounds in concentrations equivalent to the $LC_{50}$ of the EOs, that is, 16.2 μg μL$^{-1}$ (terpineol) and 25.8 μg μL$^{-1}$ (linalyl acetate). The methodology used was the same as described in Section 2.4.

The evaluation of insect mortality was carried out 12, 24, 36, 48, 60 and 72 h after application of the compounds to calculate the median lethal time ($LT_{50}$). Insects that did not move at the touch of a brush with a fine, soft tip were considered dead.

### 2.7. Life History Tables of T. absoluta Treated with LC$_{50}$ of Citrus spp. EOs

About 200 adults of *T. absoluta* were kept in an acrylic cage (60 × 30 × 30 cm) containing a tomato plant (±15 cm high) during 24 h for oviposition. After this period, the plant was removed and placed in another cage where a daily assessment was conducted with the aid of a magnifying glass (10×) to check for the appearance of larvae. The insects were evaluated daily until the second larval instar was recognized through the visualization of cephalic capsules and the size of the larvae was measured with the aid of a stereoscopic microscope (40×). After reaching the second larval instar, 445 larvae were removed with the aid of a brush with fine, soft bristles, treated with 1 μL of the $LC_{50}$ of each essential oil (*C. aurantifolia*: 33.75 μg μL$^{-1}$; *C. aurantium*: 38.78 μg μL$^{-1}$; *C. aurantium bergamia*: 35.05 μg μL$^{-1}$) with the aid of a microsyringe and individualized in the same way. The leaflets were replaced by new ones every 3 days. All plates were sealed with PVC plastic film with small holes to allow aeration and prevent the escape of insects.

The experimental design used was completely randomized, with four treatments and 115 repetitions, each one formed by a Petri dish with a treated 2nd instar larva. Larval and pupal survival, the duration of larval instars, pupal and adult stages and total development time were evaluated daily. Sexing was conducted in the pupal phase [30].

To assess the effects of EOs on surviving adults from treated second instar larvae, newly emerged couples (male and female) from each treatment were separated and maintained in the proportion of 1 couple per Petri dish (1.9 cm height × 10 cm in diameter), covered with perforated PVC plastic film to allow aeration and prevent the escape of insects. Previously, a piece of cotton wool moistened with a 10% honey aqueous solution and also a tomato leaf with 3 leaflets and the petioles fixed in moistened cotton were placed on each plate. The leaflets served as a substrate for oviposition. The number of live insects, the longevity of males and females were assessed daily. To assess the total fecundity per female and the percentage of viable eggs, the leaflets containing eggs were removed, transferred and maintained in another Petri dish after 2 days, and then, the eggs were counted and evaluated daily until larvae eclosion.

The life tables for each treatment were made using the age-stage, two-sex life table [24,26]. Biological and demographic data on life history were analyzed using the software TWOSEX-MSChart [31]. The biological parameters used were age-stage-specific survival rate (Sxj), age-specific survival rate (lx), fertility by age and stage of development (fx), age-specific fertility (mx), age-specific maternity (lxmx), age-stage life expectancy (exj), age-stage reproductive value (vxj), net reproductive rate (R0), intrinsic rate of increase (r), finite rate of increase (λ) and mean generation time (T). The means, variances and standard error of the studied parameters were compared in pairs between treatments by the bootstrap method with 100,000 replicates [26,32].

The life table considers the averages of the parameters of survival, life expectancy and fertility until the moment when it reaches age x and stage j.

*2.8. Statistical Analysis*

The data related to the insect survival over time (items 2.4, 2.5 and 2.6) were analyzed through the nonparametric estimator of Kaplan–Meier, using the Survival 3.2-7 package [33] in R software [34]. The median lethal time ($LT_{50}$) for each group formed was also estimated. The curves were compared using the pairwise test.

To determine the concentration–mortality responses of *T. absoluta* to the EOs with a 95% confidence interval, logit analysis was conducted using the DRC package [35] in R software [34]. A binomial generalized linear model (GLM) was adjusted for each EO. To determine the concentrations, the "ED" function was used in a log-logistic model with two parameters to establish the curve:

$$(Y): f(x) = \frac{1}{1 + exp(b(log\ (x) - log\ (e)\ ))}$$

where the lower limit is 0 and the upper limit is 1; *e* is the inflection point of the concentration–response curve and corresponds to the $LC_{50}$ value; *b* is proportional to the slope in the concentration *e*; and *x* corresponds to the concentration value [36]. The concentrations found were different if the confidence limits did not overlap.

The life history data, including the survival, growth, development, longevity and fertility of *T. absoluta* (item 2.6) were submitted to the analysis of the age-stage, two-sex life table using the software TWOSEX-MSchart (National Chung Hsing University, Taichung, Taiwan) [24,26]. The standard errors of the life history, reproductive and population parameters were estimated via the bootstrap technique using 100,000 resamples [32]. The differences between treatments were analyzed using the paired bootstrap test at a 5% significance level.

For the viability of eggs, the data were adjusted to a GLM with "Quasibinomial" distribution, and the averages were compared using Tukey's contrast analysis at a 95% probability using the *Multcomp* package [37] in R software [34].

## 3. Results

*3.1. Chemical Characterization of EOs*

It was possible to identify 77.18%, 97.35% and 93.05% of the compounds present in the EOs from *C. aurantifolia*, *C. aurantium* and *C. aurantium bergamia*, respectively. The monoterpene alcohol, α-terpinol, was the major compound of *C. aurantifolia* (44.74%). Linalyl acetate is a linalool acetate ester; high amounts of this component were found in EOs from *C. aurantium* (55.45%) and *C. aurantium bergamia* (58.12%) (Table 1).

**Table 1.** Chemical profile of *Citrus aurantifolia*, *Citrus aurantium* and *Citrus aurantium bergamia* using gas chromatography coupled with mass spectrometry.

| Essential Oil | RI [a] | Compound | Percentages | Method of Identification [b] |
|---|---|---|---|---|
| | 921 | tert-Butylbenzene | 2.25 | RI, GC-MS |
| | 925 | Limonene | 11.88 | RI, GC-MS |
| | 1010 | α-Fenchol | 2.41 | RI, GC-MS |
| | 1030 | 3-terpinen-1-ol | 4.5 | RI, GC-MS |
| *C. aurantifolia* | 1040 | β-Terpineol | 3.18 | RI, GC-MS |
| | 1061 | 2,3,3-Trimethyl-1,4-pentadieno | 2.2 | RI, GC-MS |
| | **1087** | **α-Terpinol \*** | **44.74** | **RI, GC-MS** |
| | 1094 | CTK1F4019 ($C_{10}H_{16}$) | 3.88 | RI, GC-MS |
| | 1404 | CTK5J8343 ($C_{12}H_{20}$) | 2.14 | RI, GC-MS |
| | | Unknown compounds | 22.82 | |

**Table 1.** *Cont.*

| Essential Oil | RI [a] | Compound | Percentages | Method of Identification [b] |
|---|---|---|---|---|
| *C. aurantium* | 998 | Linalool | 28.04 | RI, GC-MS |
| | 1087 | α-Terpineol | 6.22 | RI, GC-MS |
| | 1126 | Nerol | 0.71 | RI, GC-MS |
| | **1154** | **Linalyl acetate \*** | **55.45** | **RI, GC-MS** |
| | 1263 | Neryl acetate | 6.93 | RI, GC-MS |
| | | Unknown compounds | 2.65 | |
| *C. aurantiumbergamia* | 925 | Limonene | 4.99 | RI, GC-MS |
| | 998 | Linalool | 29.94 | RI, GC-MS |
| | **1153** | **Linalyl acetate \*** | **58.12** | **RI, GC-MS** |
| | | Unknown compounds | 6.95 | |

\* The main components of each essential oil are indicated in bold. [a] Retention index on RTX-5MS column relative to homologous series of n-alkanes. [b] Peak identification is based on RI, a comparison of retention indices with published data; GC-MS: comparison of mass spectra with those listed in NIST, Adams libraries and published data.

### 3.2. Acute Toxicity of EOs against T. absoluta

The survival of *T. absoluta* larvae was reduced after the application of EOs from *Citrus* spp. ($\chi^2$ = 177; d.f. = 3; $p \leq 0.01$) when compared to the negative control, acetone. The survival probability, 72 h after the application, was 0%, 6.67 ± 3.22% and 11.7 ± 4.14% for the EOs from *Citrus aurantifolia*, *Citrus aurantium* and *Citrus aurantium bergamia*, respectively, with overlaps between the confidence intervals. For the three EOs, the $LT_{50}$ was only 12 h (Figure 1).

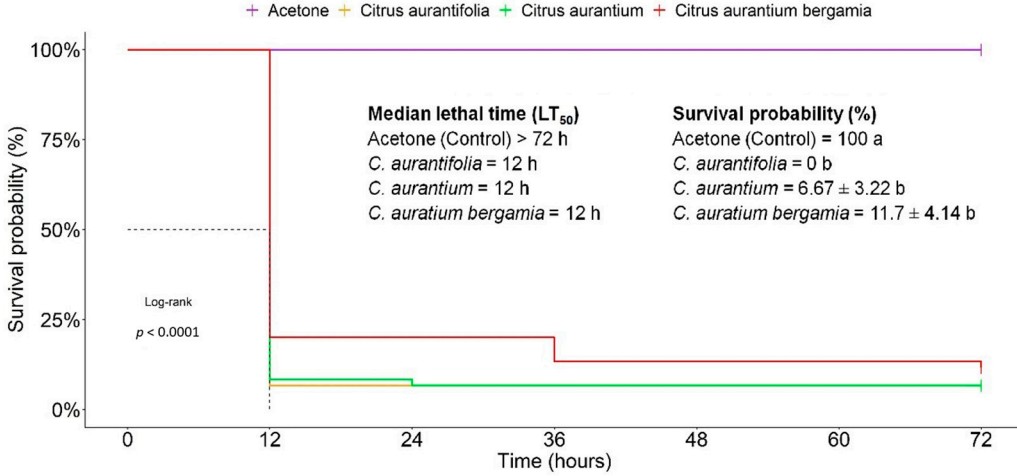

**Figure 1.** Survival curves, estimated by the Kaplan–Meier method, of *Tuta absoluta* 2° instar larvae over time submitted to topical application with essential oils from *Citrus aurantifolia*, *Citrus aurantium* and *Citrus aurantium bergamia*.

### 3.3. Time–Concentration–Mortality Responses of T. absoluta to the EOs

The larvae of *T. absoluta* treated with different doses of EOs from *C. aurantifolia* ($\chi^2$ = 167; df = 6; $p \leq 0.01$), *C. aurantium* ($\chi^2$ = 134; df = 6; $p \leq 0.01$) and *C. aurantium bergamia* ($\chi^2$ = 151; df = 6; $p \leq 0.01$) showed reduced survival rates. Larvae survival decreased as the dose of the EOs increased. Larvae treated with *C. aurantifolia* EO at doses of 42, 55, 75 and 100 μg μL$^{-1}$ presented an estimated $LT_{50}$ of $\leq$ 6 h (Figure 2A). For the *C. aurantium* EO (Figure 2B), doses greater than 55 μg μL$^{-1}$ caused mortality greater than or equal to 50% only 6 h after application. $LT_{50} \leq$ 6 h was estimated for the EO of *C. aurantium bergamia* at doses of 75 and 100 μg μL$^{-1}$ (Figure 2C).

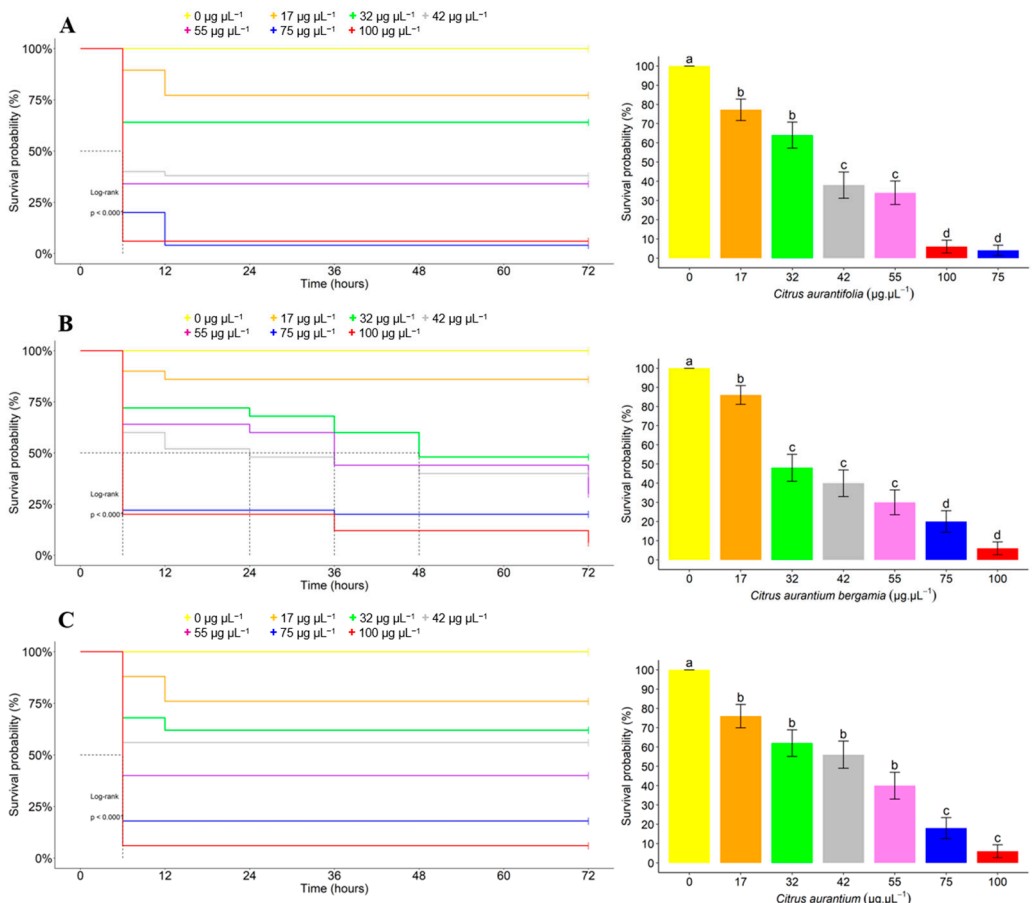

**Figure 2.** Survival curves, estimated by the Kaplan–Meier method, of *Tuta absoluta* larvae over time submitted to topical application with different concentrations of *Citrus aurantifolia* (**A**), *Citrus aurantium* (**B**) and *Citrus aurantium bergamia* (**C**). Similar lowercase letters indicate no significant differences between treatments (*p* < 0.01).

It was observed that there was an overlap between the confidence intervals of the LC$_{50}$ and LC$_{90}$ values found for the EOs from *C. aurantifolia*, *C. aurantium* and *C. aurantium bergamia*; thus, all these EOs present a similar toxicity to *T. absoluta* larvae (Table 2).

**Table 2.** Concentration–mortality responses of *Tuta absoluta* larvae to the essential oils from *Citrus aurantifolia*, *Citrus aurantium* and *Citrus aurantium bergamia*.

| Essential Oil | df | $\chi^2$ | *p* | * LC$_{50}$ | CL 95% | * LC$_{90}$ | CL 95% | *b* ** $\pm$ SE |
|---|---|---|---|---|---|---|---|---|
| *C. aurantifolia* | 10 | 12.29 | 0.27 | 33.75 | 29.31–38.18 | 88.59 | 68.77–108.42 | 2.28 $\pm$ 0.28 |
| *C. aurantium* | 10 | 10.07 | 0.43 | 38.78 | 33.62–43.94 | 113.36 | 82.41–144.31 | 2.05 $\pm$ 0.27 |
| *C. aurantium bergamia* | 10 | 4.16 | 0.94 | 35.05 | 30.43–39.68 | 94.89 | 72.46–117.33 | 2.21 $\pm$ 0.28 |

* Concentration in µg L$^{-1}$. CL: confidence limits. *b* is proportional to the slope at the LC$_{50}$ value χ2 and *p* values correspond to goodness-of-fit test ** "*b*" = coefficients of the equation f(x) = 1/1 + exp(*b*(log(x) − log(e))).

### 3.4. Acute Toxicity of Major Compounds of EOs against T. absoluta

There was a statistically significant difference between the survival over time of larvae treated with α-terpineol and linalyl acetate and the control treatment with acetone (χ$^2$ = 23.4; d.f. = 2; *p* $\leq$ 0.01). The probabilities of survival, at the end of the evaluation period of the experiment, were 100, 86.7 and 68.3% for acetone, linalyl acetate and α-terpineol, respectively. The LT$_{50}$ was greater than 72 h for all treatments evaluated (Figure 3).

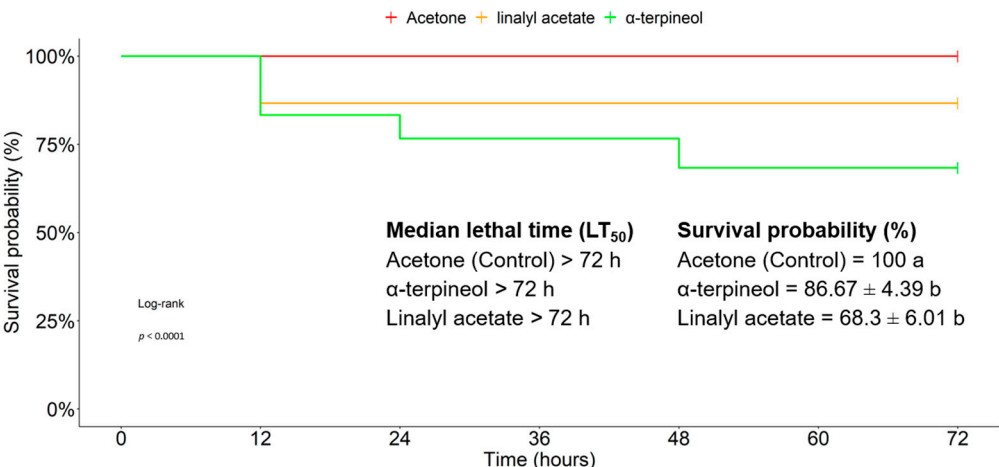

**Figure 3.** Survival curves, estimated by the Kaplan–Meier method, of *Tuta absoluta* 2° instar larvae over time submitted to topical application of major compounds of essential oils from *Citrus aurantifolia* (α-Terpinol) and linalyl acetate for *Citrus aurantium* and *Citrus aurantium bergamia*.

### 3.5. Life History Tables of T. absoluta Treated with $LC_{50}$ of Citrus spp. EOs

There were differences in the duration of the larval and pupal stages of *T. absoluta* after topical treatment with EOs from *Citrus* spp., when compared to the negative control, acetone. However, the total duration of time from egg to adult was similar to the control. Regarding the total male longevity, it was observed that insects treated with *C. aurantifolia* oil had longer longevity (Table S1).

All EOs reduced the life expectancy per stage of development of *T. absoluta* compared to control in the larval stage. In the pupal stage, the *C. aurantifolia* and *C. aurantium* EOs did not cause any negative effects, while the *C. aurantium bergamia* EO reduced life expectancy. Only the *C. aurantium* EO caused a decrease in (exj) in adult males. EOs did not reduce female life expectancy and longevity (Figure S1).

The age-stage survival rate (Sxj) indicates the probability that the insect will survive at age x and stage j. Decreases were observed in the survival of the second instar larvae of *T. absoluta* with the EOs of *C. aurantifolia* (77.39%) and *C. aurantium* (78.26%) when compared to the EO of *C. aurantium bergamia* (92.17%) and the control (90.0%). This reduction caused by *C. aurantifolia* and *C. aurantium* EOs was maintained until the third and fourth instars. In the pupal stage, the maximum value for the control treatment was 74%, while for *C. aurantifolia*, *C. aurantium* and *C. aurantium bergamia*, it was 40.87%, 36.52% and 56.52%, respectively. The EOs showed maximum averages lower than the control treatment regarding the survival of males and females (Figure 4).

The EOs of *C. aurantifolia*, *C. aurantium* and *C. aurantium bergamia* caused a decrease in the survival rate by specific age (lx) from the fifth day of life of the insects, while in the control treatment, the curve slowly decreased over time. The durations of the age-specific maternity period (lxmx) and age-specific fertility were shorter in insects treated with *C. aurantifolia* and *C. aurantium bergamia* oils (Figure S2).

The fertility of all females was reduced by the EOs. The *C. aurantifolia* EO caused a remarkable reduction in this biological characteristic when compared to the other products (Table 3). The total oviposition period was shorter for the *C. aurantifolia* and *C. aurantium bergamia* EOs, while *C. aurantium* was innocuous when compared with the control treatment. There were no negative effects of treatments in the adult pre-oviposition (APOP) and total pre-opposition (TPOP) periods. The lowest maximum daily fertility (MDF) values were observed in the *C. aurantifolia* and *C. aurantium* treatments, and the lowest maximum total fertility (MTF) was observed in the *C. aurantifolia* treatment (Table 3).

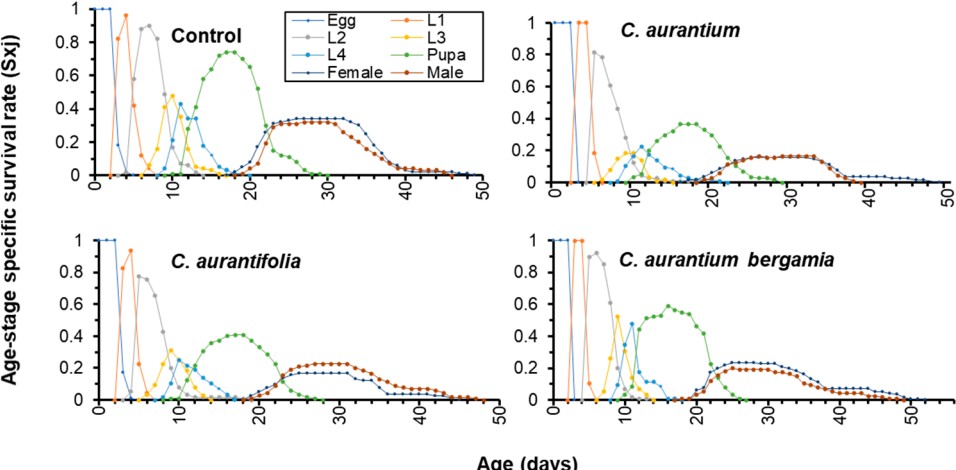

**Figure 4.** Age-stage-specific survival rate (Sxj) of *Tuta absoluta* treated with LC$_{50}$ *Citrus aurantifolia*, *Citrus aurantium* and *Citrus aurantium bergamia* EOs and only acetone (control).

**Table 3.** Reproductive parameters of *Tuta absoluta* exposed to LC$_{50}$ *Citrus aurantifolia, Citrus aurantium* and *Citrus aurantium bergamia* EOs and only acetone (control).

| Parameter | Control (Acetone) | | *Citrus aurantifolia* | | *Citrus aurantium* | | *Citrus aurantium bergamia* | |
|---|---|---|---|---|---|---|---|---|
| | N | Mean ± SE | N | Mean ± SE | N | Mean ± SE | N | Mean ± SE |
| Fecundity (E/F) total | 34 | 46.82 ± 5.76 a | 19 | 24.79 ± 4.05 b | 19 | 30.63 ± 6.1 ab | 30 | 23.1 ± 4.81 b |
| Fecundity (E/F) * | 30 | 53.07 ± 5.61 a | 16 | 29.00 ± 3.77 b | 16 | 36.38 ± 6.26 b | 18 | 38.5 ± 5.57 ab |
| Oviposition (days) | 30 | 3.43 ± 0.30 a | 16 | 2.56 ± 0.26 b | 16 | 2.81 ± 0.25 ab | 18 | 2.28 ± 0.24 b |
| PPOA (days) | 30 | 3.20 ± 0.32 a | 16 | 3.5 ± 0.52 a | 16 | 3.38 ± 0.42 a | 18 | 2.28 ± 0.4 a |
| PPOT (days) | 30 | 24.67 ± 0.37 a | 16 | 25.38 ± 0.46 a | 16 | 24.88 ± 0.54 a | 18 | 24.33 ± 0.47 a |
| MFD (E/F) | | 70 | | 30 | | 46 | | 71 |
| MFT (E/F) | | 126 | | 54 | | 85 | | 93 |

\* Total females that oviposited; PPOA (days): pre-oviposition period per adult; PPOT (days): period of total pre-oviposition; MFD (eggs/female): maximum daily fertility; MFT (eggs/female): maximum total fertility. Similar lowercase letters indicate no significant differences between treatments.

As for the age-stage reproductive value (vxj) of *T. absoluta*, there were differences in relation to the maximum values of the females, being lower in the insects treated with EOs compared with the control (Figure 5).

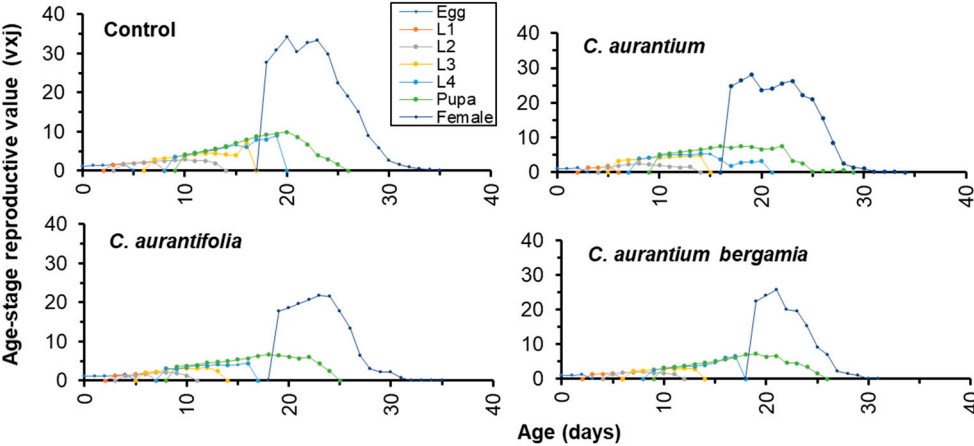

**Figure 5.** Age-stage reproductive value (vxj) of *Tuta absoluta* treated with LC$_{50}$ *Citrus aurantifolia*, *Citrus aurantium* and *Citrus aurantium bergamia* EOs and only acetone (control).

Regarding the viability of the eggs from the treated females, it was observed that all of the essential oils tested reduced the viability of the eggs ($\chi^2$ = 291; d.f. = 77; $p < 0.05$) when compared to the control treatment (Figure 6).

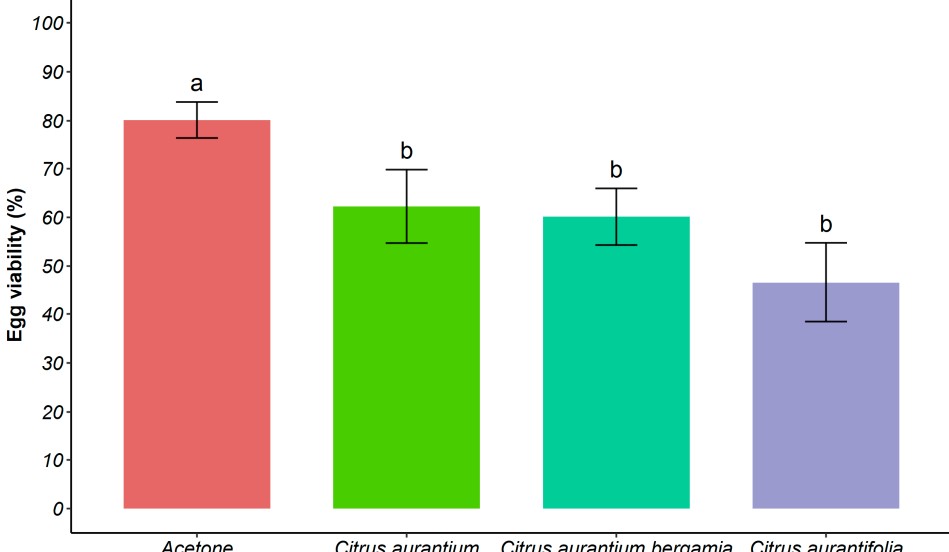

**Figure 6.** Egg viability (%) of adults of *Tuta absoluta* treated in the second larval instar with EOs $LC_{50}$ *Citrus aurantifolia*, *Citrus aurantium* and *Citrus aurantium bergamia* and only acetone (control). Similar lowercase letters indicate no significant differences between treatments.

The $LC_{50}$ of all the EOs tested caused a reduction in the intrinsic rate of increase (r), in the finite rate of increase ($\lambda$) and in the net reproductive rate ($R_0$) of *T. absoluta*. Regarding the mean generation time (T), the highest average was found in the lime EO and the lowest average in the *C. aurantium bergamia* EO (Table 4).

**Table 4.** Demographic parameters of *Tuta absoluta* treated with the $LC_{50}$ of *Citrus aurantifolia*, *Citrus aurantium* and *Citrus aurantium bergamia* EOs and only acetone (control).

| Parameter | Control | *Citrus aurantifolia* | *Citrus aurantium* | *Citrus aurantium bergamia* |
|---|---|---|---|---|
| intrinsic rate of increase (r) | 0.11 ± 0.01 a | 0.05 ± 0.01 b | 0.06 ± 0.01 b | 0.07 ± 0.01 b |
| finite rate of increase ($\lambda$) | 1.11 ± 0.01 a | 1.05 ± 0.01 b | 1.06 ± 0.01 b | 1.07 ± 0.01 b |
| net reproductive rate ($R_0$) | 15.92 ± 2.94 a | 4.10 ± 1.07 b | 5.06 ± 1.45 b | 6.03 ± 1.56 b |
| mean generation time (T) | 26.18 ± 0.48 ab | 26.83 ± 0.47 a | 26.09 ± 0.62 ab | 25.22 ± 0.47 b |

Intrinsic rate of increase ($day^{-1}$); finite rate of increase ($day^{-1}$); $R_0$, net reproductive rate (offspring per individual); mean generation time (days). Similar lowercase letters indicate no significant differences between treatments.

## 4. Discussion

In this study, it was found that the EOs of *C. aurantifolia*, *C. aurantium* and *C. aurantium bergamia* have an acute toxicity for *T. absoluta* and, at low doses, cause a reduction in the reproduction capacity of this insect. Other *Citrus*-based EOs showed a broad spectrum of actions in the physiology of insects, revealing the potential of the EOs extracted from these plants [38–41]. The most remarkable case of success is the Prev-Am®, a commercial insecticide based on a *Citrus* EO widely used against agricultural pests around the world [20]. In addition, the EOs of *Citrus* can be easily obtained due to the abundance of their raw materials at low cost, overcoming the main practical challenge that is common in the development of new biopesticides [16,17].

The $LT_{50}$ of *T. absoluta* larvae treated with *C. aurantifolia*, *C. aurantium* and *C. aurantium bergamia* was lower than 12 h. This rapid action is also possibly linked to mechanisms of action in the nervous system. This finding is in accord with studies showing that *Citrus* EO

toxicity is partially neurotoxic, acting as acetylcholinesterase (AChE) inhibitors in $Na^+/K^+$ and octopamine receptor channels [42,43].

The major compounds identified in the present work were α-terpinol in the EOs of *C. aurantifolia* and linalyl acetate in the EOs of *C. aurantium* petitgrain and *C. aurantium bergamia*. The α-terpinol is a pleasant-smelling monoterpene often found in EOs; few studies of this molecule are described in the literature, but one study reported *Liposcelis bostrychophila* Badonnel (Psocoptera: Liposcelididae) as showing a repellent characteristic of this molecule to the pest considered in the present work [44,45].

Linalyl acetate, on the other hand, is an unsaturated hydrocarbon easily oxidized when in contact with atmospheric oxygen [46]. This component was recently studied in a quantitative structure–activity relationship (QSAR) computer model to assess bioinsecticide potential, and linalyl acetate was found to have a high molecular stability and affinity to bind acetylcholinesterase (receptor), having the hardest binding to break with the lowest energy cost to the receptor, and being near to the active principles of synthetic insecticides [47]. In addition, linalyl acetate showed moderate antifeedant activity against other pests' species [48].

Both of the major compounds found in the EOs were also reported by other authors [49–52], but in low concentrations. These differences in the concentrations of certain EO components in *Citrus* spp. are the result of the methodology and structure of the plant used for extraction, the effects of the genotype used, the geographical distribution of the plant, the type of soil in which it grows, the climatic conditions it experiences, the stress level it experiences and the physiology of the plant [53]. The variation in the concentration of the major compounds in *Citrus* spp. depends on the different maturation stages [54]. Due to this natural variation, it is essential to use GC-MS to characterize the mixture of components present in studies involving botanical products such as plant extracts and EOs [15,55].

We found that the pure major compounds were not as toxic as the oils. It is necessary to consider that EOs are complex mixtures of different classes of compounds and that their toxicity may be related to either the major compound, minor compounds, the additive effect of both or the synergism of the compounds in the mixture [56]. This is because minor compounds can modulate and amplify the action of the major compounds present in EOs through changes in the physical–chemical characteristics, increasing the capacity of cell penetration, fixation and integument penetration [53,57,58]. Examples of this are molecules that, in small quantities, can inhibit enzymes such as cytochrome P450, which is responsible for metabolizing other toxic substances in the insect's body [59,60]. The synergism between chemical molecules can be used well in the management of pest resistance to synthetic and botanical active ingredients as they increase the target's physiological susceptibility to toxic molecules [60].

It was observed that the insects exposed to the $LD_{50}$ of the oils of *C. aurantifolia*, *C. aurantium* and *C. aurantium bergamia* had reduced their reproductive parameters, such as fertility and the duration of the oviposition period. The EOs decreased the variables that indicate the population growth of *T. absoluta*. The reduction in these parameters that were mediated to low doses of the EOs could possibly lead to a reduction in population size over time. The net reproduction rate (R0) indicates physiological costs related to reproductive capacity; this parameter was reduced in insects treated with *Citrus* EOs [59]. The second demographic parameter is the intrinsic rate of increase (rm) and reflects the effects of EOs on the development, reproduction and survival of the studied organisms; in this study, rm was reduced in the treated insects [61]. A third demographic parameter, the finite rate of increase (λ), is complementary to rm and reflects the growth rate in time; the treated insects also showed lower values, indicating the diminution of the sizes of future populations [62]. The last demographic parameter considered is mean generation time (T); this was less affected by the treatments, except in insects treated with *C. aurantium begarmia*, where it had lower values at the mean generation time.

Toxicological analyses that consider population parameters are more efficient in assessing the impact of the compound on the insect in prolonged periods, compared to studies

considering only the lethal effect [63]. Therefore, the use of life tables by the stage of development for two sexes allows one to more precisely determine the population changes of the pest, since it incorporates the dynamic rates of development in terms of the time and differentiation of the individual growth stages of both sexes [26]. Some authors reported similar deleterious effects of sub-doses of botanical and synthetical products, having decreased the fecundity and viability of *T. absoluta* eggs in a similar way to that found in the present study for the EOs; however, the mechanisms involved have not been fully elucidated [64].

This reduction in reproductive parameters is probably the result of perturbation in essential mechanisms, which cause physiological and behavioral changes in the insects treated; these mechanisms can be related to vitello genesis, the ovulation of mature eggs, the promotion of spermatocyte growth, the maturation of sexual organs, etc. [21]. Although there are no studies that elucidate the mechanisms of action of EOs on *T. absoluta*, some authors report that sub-doses of insecticides can render difficult the mating behavior of lepidopterans, leading to a reduction in the number of eggs produced [65,66].

In the present study, it was possible to verify that the EOs of *C. aurantifolia*, *C. aurantium* and *C. aurantium bergamia* are toxic to *T. absoluta*. In addition, when used in low concentrations, they affected the reproductive and demographic parameters of the pest. All EOs demonstrated that, in the future, they can be used to control the tomato borer *T. absoluta*. The major compounds are not the only ones responsible for the bioactivity of EOs; for this reason, future studies aiming to understand the effects of $\alpha$-terpinol and linalyl acetate when combined with other compounds in a smaller amount of the mixture will be important in order to verify the synergistic effects of the compounds. In addition, studies that seek to understand the mechanisms of action involved in the lethal and sublethal effects of *Citrus* essential oils on the life history of *T. absoluta* are required.

## 5. Conclusions

The EOs of *C. aurantifolia*, *C. aurantium* and *C. aurantium bergamia* are toxic to *T. absoluta* and, in low concentrations, the EOs reduce the fitness of the insect. In addition, the toxicity was not found to be related to the major compounds alone, which demonstrate low toxicity to *T. absoluta*.

**Supplementary Materials:** https://www.mdpi.com/article/10.3390/agriculture13030538/s1, Figure S1: Age-stage life expectancy (exj) of Tuta absoluta treated with LC50 Citrus aurantifolia, Citrus aurantium and Citrus aurantium bergamia EOs and only acetone (control); Figure S2: Age-specific survival rate (lx); Age-specific fertility (mx); Age-specific maternity (lxmx); Age-stage specific fertility (fx) of T. absoluta treated with LD50 Citrus aurantifolia (lime), Citrus aurantium (petitgrain) and Citrus aurantium bergamia (bergamot) EOs and only acetone (control); Table S1: Life history parameters of Tuta absoluta treated with essential oils from Citrus aurantifolia (33.75 µg µL$^{-1}$), Citrus aurantium (38.78 µg µL$^{-1}$) and Citrus aurantium bergamia (35.05 µg µL$^{-1}$).

**Author Contributions:** Conceptualization, G.T.d.P.S., G.A.C. and D.S.A.; methodology G.T.d.P.S., G.A.C., K.G.F., D.S.A., D.F.d.O., G.H.S., G.T.d.S.e.S., M.S.d.O. and A.B.; software, G.T.d.P.S., G.H.S. and G.T.d.S.e.S.; validation, G.T.d.P.S., G.A.C., K.G.F., D.S.A., D.F.d.O., G.H.S., G.T.d.S.e.S., M.S.d.O. and A.B.; formal analysis, G.T.d.P.S., G.A.C., K.G.F., D.S.A., D.F.d.O., G.H.S., G.T.d.S.e.S and M.S.d.O.; investigation, G.T.d.P.S. and K.G.F.; resources, G.T.d.P.S. and G.A.C.; data curation, G.T.d.P.S., G.A.C., K.G.F., D.S.A. and D.F.d.O.; writing—original draft preparation, G.T.d.P.S., G.A.C. and D.S.A.; writing—review and editing, G.T.d.P.S., G.A.C. and D.S.A.; visualization, G.T.d.P.S. and G.A.C.; supervision, G.T.d.P.S., G.A.C., K.G.F., D.S.A., D.F.d.O., G.H.S., G.T.d.S.e.S., M.S.d.O. and A.B.; project administration G.T.d.P.S. and G.A.C.; funding acquisition, G.A.C. All authors have read and agreed to the published version of the manuscript.

**Funding:** This research was funded by Fundação de Amparo à Pesquisa do Estado de Minas Gerais (FAPEMIG).

**Institutional Review Board Statement:** Not applicable.

**Informed Consent Statement:** Not applicable.

**Data Availability Statement:** The data presented in this study are available on request from the corresponding author.

**Conflicts of Interest:** The authors declare no conflict of interest. The funders had no role in the design of the study; in the collection, analyses, or interpretation of data; in the writing of the manuscript; or in the decision to publish the results.

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
