# Peer review of "Survival and Demography of the Tomato Borer (Tuta absoluta) Exposed to Citrus Essential Oils and Major Compounds"

_agriculture, doi:10.3390/agriculture13030538_

Round 1

Reviewer 1 Report

Comments and Suggestions for Authors

The manuscript ‘Survival and demography of the tomato borer (Tuta absoluta) exposed to Citrus essential oils and major compounds’ reported the application value of Citrus essential oils in the control of Tuta absoluta. This pest is small in body size and can be easily carried away via national and international trade activities of agricultural products, especially the tomato fruit, posing a potential threat to the safety production of agricultural industry. However, the manuscript is still immature. The list of corrections described below, but especially the introduction, should be reconsidered.

LINE 11 The abstract of this article is a bit weak and the details should be provided.

LINE 41-43 Please provide the citation.

LINE 63 There are many reports on essential oils in the control of Tuta absoluta. Please add the status of the research.

LINE 48-50 Please add data on economic losses caused by Tuta absoluta, especially to the world's largest tomato producer.

LINE 80 Why did the authors choose citrus essential oils? Please add in the introduction.

LINE 93 Please provide the specific values for greenhouse conditions.

LINE 94 Please state the model and manufacturer of the vacuum pump.

LINE 100 Are the essential oils commercial? Please state the purity of the essential oils.

LINE 349 Please state the citation.

LINE 345 Are the better concentrations from the results of this study applicable to actual pest control? How should they be used? Please discuss.

LINE 348-354 Put a preface in the introduction. 

LINE 346 The conclusion should be added as a separate paragraph in my opinion.

Author Response

LINE 11 The abstract of this article is a bit weak and the details should be provided

Some details were added, mostly related to the major compounds bioassay that were not present in the abstract.

LINE 41-43 Please provide the citation.

checked

LINE 63 There are many reports on essential oils in the control of Tuta absoluta. Please add the status of the research.

Checked

LINE 48-50 Please add data on economic losses caused by Tuta absoluta, especially to the world's largest tomato producer.

Checked

LINE 80 Why did the authors choose citrus essential oils? Please add in the introduction.

Checked

LINE 93 Please provide the specific values for greenhouse conditions.

Checked

LINE 94 Please state the model and manufacturer of the vacuum pump.

Checked

LINE 100 Are the essential oils commercial? Please state the purity of the essential oils.

Checked

LINE 349 Please state the citation.

Checked

LINE 345 Are the better concentrations from the results of this study applicable to actual pest control? How should they be used? Please discuss.

This study does not have the objective of generating technology to be apllied imediately in the field, as it is classified as basic research and has an exploratory character. At first, it is not possible to use these Citrus pure essential oils and major components, requiring further work that focuses on generating formulations that stabilize and protect molecules contained in EOs. Such studies will require basic research, such as the present study.

LINE 348-354 Put a preface in the introduction. 

Checked

LINE 346 The conclusion should be added as a separate paragraph in my opinion.

Checked

Reviewer 2 Report

Dear Editor and authors,

Silva et al. present interesting laboratory results evaluating the toxicity of three essential oils from Lime, petitgrain and bergamot, and they provide great deal of data.  Overall, the MS is well written and organised. I recommend it for publication, but I suggest that authors consider some issues and seek for English editing.

My major issue is that the authors used laboratory rearing of Tuta absoluta (L89). Its well know that laboratory insects are more sensible to insecticides than ones from the field. Thus, how long the insect has been reared in the laboratory? Do the authors renew the rearing?

In discussion, its important to highlight that ther results only based on laboratory tests and further experiments need to be done in field/cage experiments.

L52-56: make it in one sentence. To show that inefficiency comes from insecticide resistance.

L61: provide the common name.

L101-105: If I understand well, the EOs come from commercial company. If so, why you used gas chromatograph coupled to a mass spectrometer?

L138:Complet name when starting a sentence.

L149: the sentence is not complete.

L233: rephrase

L244: The Value of (0) belongs to what?

L376: delete “it”.

L383: which study

L415-416: delete

L417-419: rephrase

L420-421: are you certain? Delete

L421-423: what kind of mechanisms!!

I did not edit the supplementary materials because tje documents could not be opened.

Author Response

L52-56: make it in one sentence. To show that inefficiency comes from insecticide resistance.

Checked

L61: provide the common name.

Checked

L101-105: If I understand well, the EOs come from commercial company. If so, why you used gas chromatograph coupled to a mass spectrometer?

Generally, essential oils may present variations in their composition due to several factors such as the region where the material was collected, age and physiological state of the plant and tissue used, season of collection, among others. In order to confirm the exact composition of the product, we prefer to carry out the analysis.

L138:Complet name when starting a sentence.

checked

L149: the sentence is not complete.

checked

L233: rephrase

checked

L244: The Value of (0) belongs to what?

checked, this is the mean value of probability survival in the firt reported treatment (Citrus aurantifolia). 0%

L376: delete “it”.

checked

L383: which study

checked

L415-416: delete

checked

L417-419: rephrase

checked

L420-421: are you certain? Delete

checked

L421-423: what kind of mechanisms!!

checked